# Supervised Machine Learning Methods for Seasonal Influenza Diagnosis

**DOI:** 10.3390/diagnostics13213352

**Published:** 2023-10-31

**Authors:** Edna Marquez, Eira Valeria Barrón-Palma, Katya Rodríguez, Jesus Savage, Ana Laura Sanchez-Sandoval

**Affiliations:** 1Genomic Medicine Department, General Hospital of México “Dr. Eduardo Liceaga”, Mexico City 06726, Mexico; valeirabarron@gmail.com (E.V.B.-P.);; 2Institute for Research in Applied Mathematics and Systems, National Autonomous University of Mexico, Mexico City 04510, Mexico; katya.rodriguez@iimas.unam.mx; 3Signal Processing Department, Engineering School, National Autonomous University of Mexico, Mexico City 04510, Mexico; robotssavage@gmail.com

**Keywords:** machine learning, decision support system, medical diagnosis, influenza, artificial intelligence

## Abstract

Influenza has been a stationary disease in Mexico since 2009, and this causes a high cost for the national public health system, including its detection using RT-qPCR tests, treatments, and absenteeism in the workplace. Despite influenza’s relevance, the main clinical features to detect the disease defined by international institutions like the World Health Organization (WHO) and the United States Centers for Disease Control and Prevention (CDC) do not follow the same pattern in all populations. The aim of this work is to find a machine learning method to facilitate decision making in the clinical differentiation between positive and negative influenza patients, based on their symptoms and demographic features. The research sample consisted of 15480 records, including clinical and demographic data of patients with a positive/negative RT-qPCR influenza tests, from 2010 to 2020 in the public healthcare institutions of Mexico City. The performance of the methods for classifying influenza cases were evaluated with indices like accuracy, specificity, sensitivity, precision, the f1-measure and the area under the curve (AUC). Results indicate that random forest and bagging classifiers were the best supervised methods; they showed promise in supporting clinical diagnosis, especially in places where performing molecular tests might be challenging or not feasible.

## 1. Introduction

Influenza is a respiratory disease that can increase the incidence of pneumonia and cause a high number of hospitalizations [1]. In March 2009, Mexico, the United States and Canada were the focus of international attention when the influenza A H1N1 virus burst onto the epidemiological scene [2]. In June of that same year, the World Health Organization (WHO) declared an influenza pandemic of moderate severity. Since 2009, respiratory diseases due to influenza have recurred in numerous nations during the colder months annually, thus acquiring the category of seasonal influenza. There are four types of influenza viruses: A, B, C and D. Influenza A and B viruses cause seasonal epidemics of disease, and have been responsible for thousands of deaths worldwide, despite the annual vaccination campaigns [3]. In Mexico, between 2020 and 2021, the incidence of influenza decreased substantially due to the Coronavirus disease (COVID-19) caused by the Severe Acute Respiratory Syndrome Coronavirus 2 (SARS-CoV-2); however, influenza cases increased again in 2022 [4]. Before the COVID-19 pandemic, an estimated 291,000 to 646,000 respiratory deaths occurred worldwide each year due to seasonal influenza [5,6]. In Mexico, information regarding influenza infections has been registered since 2010 in the Influenza Epidemiological Surveillance System to identify its behavior and be able to predict how the next influenza season will develop [7]. The data in this system are obtained from symptomatic patients treated at healthcare centers and who had undergone a quantitative reverse transcription polymerase chain reaction (RT-qPCR) test to detect the presence of influenza viral RNA.

The Centers for Disease Control and Prevention (CDC) specify the common symptoms experienced by influenza patients, such as fever or feeling feverish/chills, cough, sore throat, runny or stuffy nose, muscle or body aches, headaches, and fatigue (tiredness) [8]. The presence of these symptoms is not a guarantee of having been infected by the influenza virus; moreover, they vary among the population. Distinguishing the causal agent of the illness between the influenza virus and other viral or bacterial agents proves to be challenging through clinical evaluation alone. Therefore, other tests should be applied to confirm the diagnosis of influenza, RT-qPCR being the most successful test for the molecular diagnosis [9]. However, in developing countries, this test is not routinely performed due to high costs and the limited availability of testing facilities. Not all hospitals and clinics have the necessary equipment and supplies to perform these tests, leading to potentially lengthy turnaround times [10].

We propose to use alternative methods to facilitate the diagnosis of influenza, like methods based on Artificial Intelligence (AI), as they could serve as tools to assist in medical attention for diagnosis prior to RT-qPCR tests or can be applied in locations with difficult access to molecular analysis. In Figure 1, the workflow applied in this work to find the best Machine Learning method to diagnose influenza is shown.

Machine Learning (ML) is a component of AI that encompasses a set of techniques which enable the implementation of adaptive algorithms to make predictions and self-organize input data based on common characteristics. When ML is trained with a correct data set and the algorithm is standardized to accurately respond to all inputs, it is called a supervised ML [11].

Since the 1970s, interest in applying AI-ML in the health sector has grown [12]. In the medical field, a significant amount of patient data is managed, including sociodemographic and epidemiological information, results from physical examinations, diagnostic support test outcomes, and procedures performed, among others [10,13,14,15,16,17,18,19,20,21,22,23,24,25,26,27,28,29,30,31,32,33,34,35,36,37,38,39,40,41,42,43,44,45,46,47]. Because ML can effectively handle a large number of attributes (features), and due to its ability to identify and leverage the interactions among these numerous attributes, it becomes a particularly compelling tool in this domain [20]. ML algorithms have found extensive application across numerous medical specialties, serving purposes in prevention, diagnosis, treatment, and survival analysis alike.

In the case of influenza, ML has been applied to achieve several objectives, one of which is predicting the incidence of cases in the upcoming influenza seasons [38,39,40], including predicting the most prevalent types of influenza viruses for the season [41,42,43]. In the diagnostic stage, studies have demonstrated how metabolomic data from patients can be used to infer whether they are positive or negative for influenza [44]. There is even a report in which open access data were employed to develop a classifier for influenza diagnosis; however, not all included patients had a RT-qPCR result to confirm the diagnosis and validate the classifier’s functionality [10]. ML has also been applied to forecast the efficacy of influenza vaccines [45,46,47].

## 2. Materials and Methods

### 2.1. Data Set

In this study, a clinical data set comprising 19,160 patient records from Mexico City was used. The database was made to track influenza’s seasonal behavior and make prognosis for the next season. Data excluded patients’ names, home addresses and hospital registration numbers. The data set was exported to the authorized researcher for this retrospective study, which was reviewed and approved by the institutional ethical committee (D1/19/501-T/03/096).

We applied three exclusion criteria: (1) age < 7 years, (2) patients with a negative influenza test but positive for another respiratory virus, and (3) no RT-qPCR result record. After these criteria, the study included 15,480 records of patients aged between 7 and 119 years old. Figure 2 shows the data distribution. The age ranges were 7–19 (41.5%), 20–39 (20.1%), 40–59 (21.2%), and age ≥ 60 (17.2%); according to the RT-qPCR test, 11,268 (72.8%) were negative and 4212 (27.2%) were positive for influenza virus, and the distribution by sex was 7710 (49.8%) men, and 7770 (50.2%) women. The data set consisted of 24 attributes encompassing clinical and demographic information collected from patients upon arrival at healthcare institutions for clinical examination and before the sample taking (nasopharyngeal and/or oropharyngeal exudate) for a RT-qPCR test.

The set of symptoms and demographic features selected was the vector. The data were subjected to manual labeling by a clinician who assessed each patient and assigned values of ‘yes’ or ‘no’ to denote the presence or absence of symptoms, and ‘unknown’ otherwise.

### 2.2. Data Preprocessing

The symptoms and demographic data were labeled by binary numbers to indicate the presence or absence of a feature (1, 0, respectively); an unknown case was labeled as 0. Age range was mapped into the range {0…1}, to normalize the data.

In this study, two-step approaches were used to select the main features for training and testing the supervised ML methods. In the first step, Spearman’s correlation was used to determine the correlation coefficient between features, as these are categorical variables. We selected them with a weak correlation (r < 0.75). In the second approach, chi-squared was computed to analyze the association between independent variables and influenza. The features selected had a strong association (*p* < 0.001). 

To examine ML models and evaluate their performance, the data set was randomly split into 80% for the training set and 20% for the testing set. The models were evaluated with 10-fold cross-validation to select the best one. Python 3 functions were applied to create the k-Fold distribution and stratification.

The original data set was unbalanced, with the majority of the cases being negative for influenza (72.8%) and a minority of cases with a positive influenza test (27.2%). In this study, the target classes of the training set were balanced 50:50, and the skew was eliminated to obtain a most appropriate performance of the ML methods [48]. The minority oversampling technique used was Random Oversampling (ROS), which increases the size of the data set by randomly resampling the original minority class without creating new samples or changing the sample variability [49]. All samples from the majority class (negative for influenza) were used, and through ROS, data were added to the minority class (positive for influenza), obtaining an equal number of samples in both classes. 

### 2.3. Machine Learning Algorithms

ML methods are an automatic and objective way to classify the samples into two classes, positive or negative for influenza, using records with inputs and outputs for the process, features and their classification [50,51,52]. In this way, we tried to find patterns in the known data in order to apply them to the new unclassified data.

We had the pair (X,Y) in all cases, X = {x_1_, x_2_, …, x_n_} and Y = [positive|negative] for influenza, according to the RT-qPCR test. 

The set of signs, symptoms and demographic features selected are represented by the vector X, and the data are binary numbers that indicate the absence or presence of the feature. The age was normalized to a range {0…1}. 

The aim was to find a model *F* of ML to represent the approximation between inputs and outputs.
*F*: X → Y(1)

For this study, 10 popular supervised ML methods through python.sklearn libraries for binary classification were used [53,54]: Adda Boost, Decision Tree, Bagging classification, Gradient Boosting Classifier (GBC), Random Forest (RF), K-nearest neighbors (Neighbors KNN), Naive Bayes (NB), Support Vector Machine (SVM), Logistic Regression (LR) and Discriminant Analysis. Taking advantage of the implementation of supervised ML models available in the python 3.7 programming language, the tests were carried out with several algorithms to evaluate their performance and be able to select the most accurate for this case study, and not only with the classical ML algorithms such as SVM, decision tree, KNN and NB.

Adaptive Boosting, named the AdaBoost algorithm, is a ML Meta-algorithm that can be used with many other ML algorithms to enhance its performance [55]. AdaBoost is sometimes denominated as the strongest out-of-the-box classifier for the so-called weak learners [56].

Decision trees used to predict categorical variables are called classification trees and decision trees [49]. The decision tree classifier is a flowchart-like tree structure; each internal node represents a test on an attribute, each branch represents an outcome of the test, and the class label is represented by each leaf nodes (or terminal nodes). Decision trees can be transformed into classification rules [57]. This ML algorithm is used to create the ensemble ML methods.

Random forest is a set of decision tree classifiers; in this ML model, each decision tree depends on a random vector of the training data set. They vote independently for the most popular class, and their classification is ensembled to give the final output using the classes given by each model [58]. The random forest algorithm is a special type of ensemble method. A random forest consists of many small classification decision or regression trees. Each tree, individually, is a weak learner; however, all the decision trees together can build a strong learner. It is random because (a) when building trees, a random sampling of training data sets is followed; and (b) when splitting nodes, a random subset of features is considered [59].

Bagging is a classifier which generates different subsets of the training data set by selecting data points randomly and with replacement. It can select the same instance multiple times. It is also called bootstrap aggregation and was created before the random forest. Given that a small change in the data can bring diverse effects in the model, the structure of the tree can completely change each tree to randomly sample the data set with a replacement, results on different trees [60]. 

The random forest algorithm is considered an extension of the bagging method. The difference is the number of features used in the decision tree construction: in bagging, all attributes are used for every decision tree, whereas in random forest, the decision trees have a random sample of attributes. Both ML methods are based on the decision tree method. The decision tree method works only with one tree to represent all samples and can be overfitting. Bagging and random forest work with several trees to represent different types of samples for each one.

Gradient boosting is a class of ensemble algorithms for machine learning that is used for regression or classification prediction modeling problems. It combines several sequential classifiers [61]. At an iteration, trees are added to the ensemble to fit correctly to the prediction errors made by prior models (boosting) and model fittings, using any arbitrary differentiable loss function and gradient descent optimization algorithm. The techniques is known as gradient boosting (Gboost) [62].

In the case of the KNN classifier, the main goal is to predict the closest value using distance as a basis. The Euclidean distance is a widely used technique [48]. The classification of the input data is based mainly on the selection of the majority class among its nearest neighbors [63].

The Naive Bayes classifier is considered a powerful probabilistic algorithm, based on Bayes theorem: the word “naïve” indicates interdependencies between characteristics. The version Bernoulli Naive Bayes (BNB) is used for Boolean variables as predictors [64,65].

Support Vector Machine (SVM) is a popular machine learning tool, which offers solutions to problems in classification and regression [66]. SVM separates classes and finds the hyperplane that best separates the data into different classes with a maximal margin between the classes. Initially using a linear decision boundary called a hyperplane to classify the data, Vapnik introduced a way of building a nonlinear classifier by using kernel functions [67]; it is placed at a location that maximizes the distance between the hyperplane and instances [68].

On the other hand, Discriminant Analysis aims to classify objects, by a set of independent variables, into one of two or more mutually exclusive and exhaustive categories. Discriminant analysis can be used only for classification (i.e., with a categorical target variable), not for regression. The target variable may have two or more categorical data by the use of multivariate information from the samples studied [69]. There are two methods: linear and quadratic discrimination. The first is the most widely used where classes are linearly separated. When a multi-classes analysis is needed, the two-groups method is used repeatedly in the analysis of pairs of data and the separation is linearity. Quadratic discrimination is used with nonlinearly separable classes.

### 2.4. Validation 

The model performances were evaluated with a k-fold cross-validation method, which is an objective way to find the most robust ML algorithm, and we used the contingency table with the classification results [70].

In k-fold cross validation, the entire data set was divided into k equal parts, with k-1 subsets used for training while the remaining set was for testing. Each algorithm was trained and tested k times and the model output for each sample configuration obtained using cross validation was averaged to provide the global performance output of the model. The partition of the set with the folds in the subsets was arbitrary and with equal numbers of positive and negative cases in the training. 

The confusion matrix helps to compare the classification result, and it has 4 values: true positives (TP), false positives (FP), true negatives (TN), and false negatives (FN). The columns in the matrix are the results obtained, while in the rows are the expected and results.
$$\left(\begin{array}{cc} \mathrm{TP} & \mathrm{FP}\\ \mathrm{FN} & \mathrm{TF} \end{array}\right)$$

We compared the performance of the methods in cross validation with the results using the following metrics:(2)$$Accuracy,Acc=\frac{TP+TF}{TN+FP+FN+TP}$$
This measure is for the samples correctly classified.
(3)$$Precision,Prec=\frac{TP}{TP+FP}$$
measures the positive samples correctly classified vs. only positive samples.
(4)$$Recall,Rec=\frac{TP}{TP+FN}$$
calculates the positive samples correctly classified vs. the samples expected to be positive. This is also called sensitivity.
(5)$$Specificity,Spec=\frac{TN}{TN+FP}$$
measures the fraction of negative samples classified as negative.
(6)$$F1\text{-}score,F1=\frac{2TP}{2TP+FP+FN}$$

It is an average between recall and precision.

Additionally, all the models were evaluated with the area under the curve (AUC) score in ROC analysis. This measure uses the ROC curve that shows the ability to make the difference between 2 classes with a graphical method using recall and specificity, and the AUC summarizes the performance of the classifiers in the training.

## 3. Results

At the beginning, the database had 24 attributes. According to Spearman’s correlation, the feature arthralgia was highly correlated with myalgia (r = 0.87); hence, only myalgia was selected in the analysis, as it had fewer missing values. In the chi-squared test (*p*-value in Table 1), factors like sex, diarrhea and vaccination presented a low association with influenza (*p* > 0.01); therefore, these factors were also dismissed. Finally, 20 features were selected, many more than the 8 main symptoms of influenza indicated by the CDC: fever or feeling feverish/chills, cough, sore throat, runny or stuffy nose, muscle or body aches, headaches, fatigue (tiredness). Age data were used like one factor normalized in the range {0…1}.

The number of samples to test with ML was 15,840. This database was unbalanced with 11,268 (72.8%) negative and 4212 (27.2%) positive for influenza, according to the RT-qPCR tests. In this work, Random Over Sampling (ROS) was used to increase the number of positive records in the training set to improve the method performance with an equal number of samples in positive and negative classes.

The features sex, vaccination and diarrhea were not significant in the chi-squared test, with the dependent variable influenza: these features showed small variation between the positive and the negative classes.

Finally, our balanced training set had 7918 of positive and the same number of negative rows and 20 columns of features for training and testing 10 supervised ML algorithms, validated with k-fold cross validation (k = 10). In Table 2 are the results of ML methods with cross-validation. RF had the best evaluation (AUC = 0.94, Acc = 0.86, Rec = 0.91 and Spec = 0.88 were the best); in second place was the bagging classifier, which works similarly to RF. With the resampling technique, the number of the minor class (positive for influenza) increased, and the scores reflected the equilibrium. Figure 3 shows the ROC curves of the four best ML methods in the 10-fold cross-validation. 

Random Forest, Bagging, and Decision Tree are supervised learning algorithms that can effectively handle categorical or binary features like the ones we have. The applied resampling seemed to have benefited the sensitivity results as it increased the number of positive samples in the training set. Bagging and Random Forest are ensemble techniques, based on Decision Tree, which ranked third, and along with Kneighbors, are algorithms capable of capturing non-linear relationships in the data.

The performance of the top four ML methods is shown in Figure 4, and their respective ROC plots are shown in Figure 5, both showing the results obtained with the test set. The RF and bagging methods demonstrated the highest scores when applied to the independent samples in the test set. However, it is important to note a substantial disparity between the sensitivity and specificity results in the test set. random forest achieved a sensitivity (rec) of 0.30 and a specificity (spec) of 0.90, while bagging had a sensitivity of 0.29 and a specificity of 0.88. Additionally, the significant differences observed between the results during the training and testing phases highlight certain limitations in this study. One potential factor contributing to these limitations is the sensitivity of machine learning models to class imbalance. Even after implementing random oversampling (ROS) on the training set to address the issue of imbalance, especially considering the considerably higher proportion of negative samples compared to positive samples, the desired variability was not effectively introduced into the training data. Another contributing factor might be attributed to the nature of the data set itself. In this study, we utilized binary data to represent the presence or absence of specific characteristics, with the exception of age, which was represented as a continuous variable. In contrast, other studies in the medical field that have achieved superior results have not only incorporated binary variables but have also integrated continuous variables obtained from laboratory tests and biometric measurements to assess patients’ conditions [19,20,32,37]. An illustrative case from [21] involved the transformation of continuous data into binary values, but this approach also yielded unsatisfactory results. It is plausible that, in our case, the lack of relevant information and the use of subjective values to evaluate the health status of patients may have led to weaker associations between these characteristics and the occurrence of influenza.

## 4. Discussion

The use of artificial intelligence techniques through ML has increased its possibility as an alternative or powered tool in the diagnosis of infectious diseases [71,72,73]. With this idea, in this work we searched for an ML method as an alternative to the PCR test to perform diagnostics of influenza. Here, 19 category features and age were used, collected when the patient arrived with symptoms of influenza. In order to eliminate the unbalanced data between the positive and negative influenza cases of 15840 samples and to reduce the skew, we applied a resampling technique, random over sampling to positives. With resampling techniques, the Ensemble ML methods like Random Forest and Bagging could be favored—RF with AUC = 0.94, Acc = 0.86, Spec = 0.88 and, sensibility = 0.91, and Bagging with AUC = 0.93, Acc = 0.85, Spec = 0.87 and sensibility = 0.90—in the cross-validation. In other works [74,75] RF and Bagging were also combined with resampling techniques and showed good performance. 

Nevertheless, it is important to note that these ML methods may serve as valuable screening tools to assist medical practitioners in distinguishing between positive and negative influenza cases, yielding promising results that could aid in decision making. This is particularly relevant for scenarios where the RT-qPCR test results are expected to be negative, potentially leading to reduced costs associated with testing.

Our research findings were based exclusively on data obtained from Mexican patients. This approach was chosen due to the unique health conditions prevalent in Mexico, which may differ significantly from those in other countries. It is important to consider that COVID-19 has changed the patterns of respiratory diseases [76,77]. Even though vaccines are applied every year, many people around the world are infected with influenza, causing a large number of deaths [4,78,79]. 

The potential advantage highlighted in our study is the use of an alternative decision-support tool, particularly relevant to regions where healthcare providers or patients, armed solely with basic questioning information, can assess the necessity for treatment and the conduction of PCR tests for the influenza disease.

### 4.1. Limitations of Work

Our results show problems in the prediction of positive influenza cases, maybe because the data set is imbalanced, and the binary features lose representativeness of the patient’s health status. ML techniques hold potential in diverse applications; however, it is crucial to acknowledge that, in this study, these methods play a limited role as detection tools. They should not be perceived as a complete substitute for clinical diagnosis. RT-qPCR tests retain their indispensable status for precise influenza results, and therefore, machine learning models should be considered as complementary rather than as a complete replacement for conventional diagnostic approaches. Our findings indicate challenges in predicting positive influenza cases, possibly due to data imbalance and the diminished representativeness of binary features concerning a patients’ health status. 

It is possible that the low prediction values could be improved with our data set through several avenues. One approach could involve grouping individuals according to age, as symptoms may exhibit more pronounced patterns within specific age groups. Exploring alternative machine learning models is also a worthwhile consideration in our quest for improved predictions. Additionally, expanding the data set by including more positive cases from different Mexican regions could enhance the models’ performance.

### 4.2. Future Work

This research has the potential for ongoing improvement and broader application. Comprehensive ablation studies can provide deeper insights into the algorithm’s capabilities, allowing for a clearer grasp of its strengths and weaknesses. These studies encompass various facets, including feature selection, the incorporation of additional continuous data to enhance patient health assessment, the adoption of class balancing techniques, and the use of advanced machine learning models like convolutional neural networks to handle larger data sets and continuous data. Furthermore, it is essential to explore the creation of comprehensive models that effectively differentiate between COVID-19 and influenza cases.

## 5. Conclusions

In this study, machine learning models showcased a notably higher specificity compared to sensitivity, suggesting their potential utility in the identification of negative cases. This capability could help minimize the number of unnecessary molecular tests for individuals presenting with symptoms resembling influenza. This aspect is particularly pertinent in Mexico, where, for epidemiological reasons, during the influenza season around 10% of the population with symptoms resembling influenza are randomly selected for RT-qPCR testing, with approximately 70% of those cases turning out to be negative. By incorporating a tool akin to the one outlined in this study, clinicians can make more informed decisions about which patients require PCR testing, ultimately enhancing data quality for national-level decision making. Furthermore, given the limited availability of RT-qPCR testing facilities in certain areas, this tool can serve as valuable support for healthcare practitioners, aiding them in determining the necessity of conducting tests. This approach has the potential to reduce costs for patients and ease the burden on the healthcare sector.

## Figures and Tables

**Figure 1 diagnostics-13-03352-f001:**
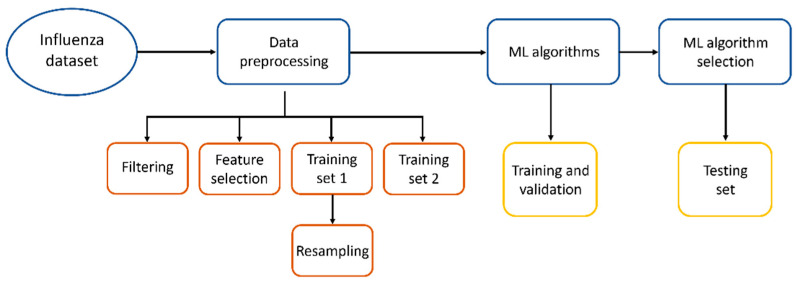
Workflow to find the best ML method to diagnose influenza.

**Figure 2 diagnostics-13-03352-f002:**
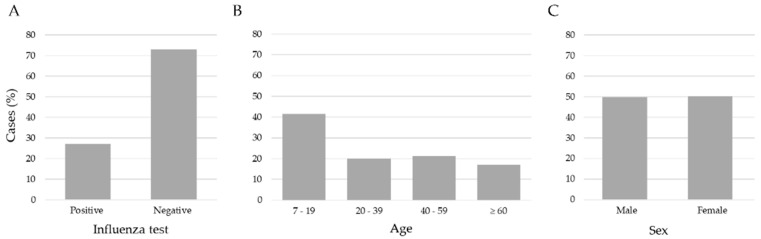
Distribution of influenza cases, ages and sex in the sample of 15,480 patients. (**A**) The number of negative (72.8) and positive (27.2) cases of influenza, (**B**) the ages separated into age ranges; (**C**) the % of feminine (50.2) and masculine (49.8) samples were very similar.

**Figure 3 diagnostics-13-03352-f003:**
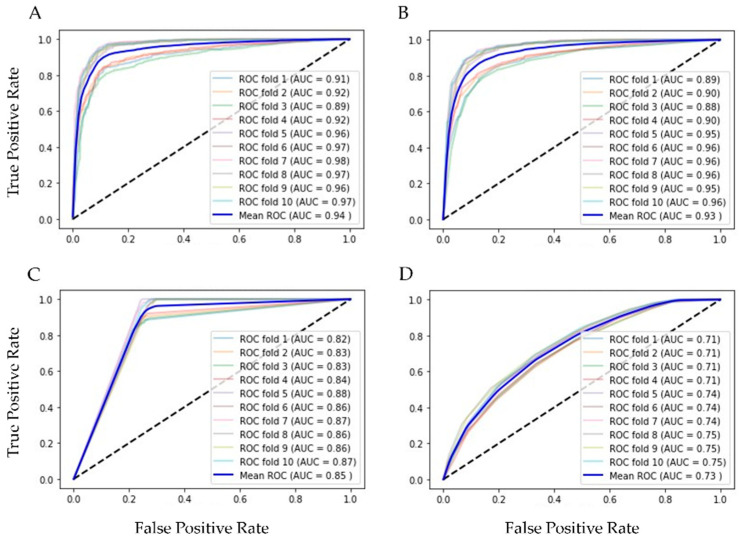
ROC graphics of the best ML algorithms with training set. (**A**) Random Forest, AUC = 0.94 ± 0.004; (**B**) Bagging, AUC = 0.93 ± 0.004; (**C**) Decision Tree, AUC = 0.85 ± 0.006; and (**D**) Kneighbors (7), AUC = 0.73 ± 0.014.

**Figure 4 diagnostics-13-03352-f004:**
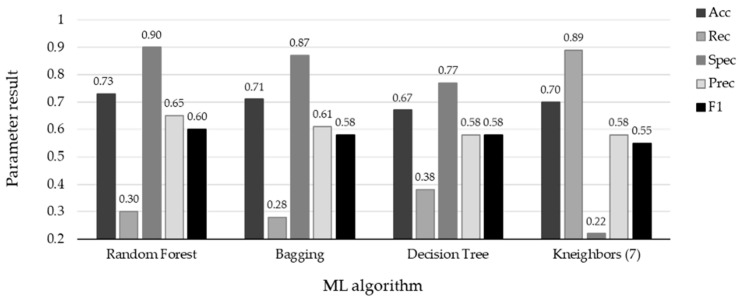
Results with test set using the four ML algorithms. The top scores are associated with specificity for the four algorithms: RF (spec = 0.90), Bagging (spec = 0.87), DT (spec = 0.77), and KNN (spec = 0.89). Conversely, the remaining metrics indicate the misclassification of positive influenza cases.

**Figure 5 diagnostics-13-03352-f005:**
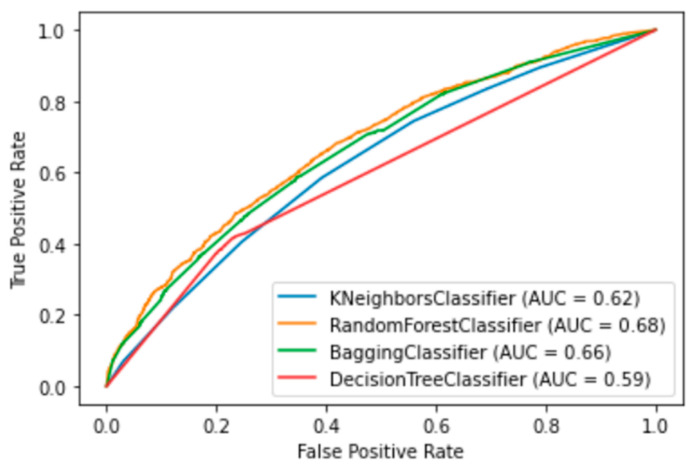
ROC graphics of the four best ML algorithms. Random Forest, Bagging, Decision Tree and Kneighbors (7).

**Table 1 diagnostics-13-03352-t001:** Attributes from influenza database of Mexico City.

Attributes	All Patients *n* = 15,480 *n* (%)	Positive *n* = 4212 *n* (%)	Negative *n* = 11,268 *n* (%)	*p*-Value
*Demographic information*				
Sex—Feminine	7770 (50.2)	2162 (51.3)	5608 (49.8)	0.087
Sex—Masculine	7710 (49.8)	2050 (48.7)	5660 (50.2)	
Hospitalized	10516 (67.9)	2449 (58.1)	8067 (71.6)	<0.001
Contact influenza-patients	2012 (13.0)	715 (17.0)	1297 (11.5)	<0.001
Vaccinated for influenza	2096 (13.5)	534 (12.7)	1562 (13.9)	0.059
Age 7–19 years	6417 (41.5)	1511 (35.9)	4906 (43.5)	<0.001
Age 20–39 years	3111 (20.1)	967 (23.0)	2144 (19.0)	
Age 40–59 years	3283 (21.2)	1056 (25.0)	2227 (19.8)	
Age ≥ 60 years	2669 (17.2)	678 (16.1)	1991 (17.7)	
*Symptoms*				
Fever	13,112 (84.7)	3853 (84.2)	9259 (82.2)	<0.001
Cough	13,953 (90.1)	3918 (85.7)	10,035 (89.1)	<0.001
Chest pain	3750 (24.2)	1160 (25.4)	2590 (23.0)	<0.001
Dyspnea	8642 (55.8)	2079 (45.5)	6563 (58.2)	<0.001
Irritability	4688 (30.3)	1159 (25.3)	3529 (31.3)	<0.001
Diarrhea	1833 (11.8)	492 (10.8)	1341 (11.9)	0.727
Shaking chills	5738 (37.1)	2003 (43.8)	3735 (33.1)	<0.001
Headache	8692 (56.1)	2896 (63.3)	5796 (51.4)	<0.001
Myalgia	6255 (40.4)	2279 (49.8)	3976 (35.3)	<0.001
Arthralgia	5539 (35.8)	2014 (44.0)	3525 (31.3)	<0.001
Malaise	9826 (63.5)	2947 (64.4)	6879 (61.0)	<0.001
Rhinorrhea	9277 (59.9)	2817 (61.6)	6460 (57.3)	<0.001
Polypnea	4602 (29.7)	1073 (23.5)	3529 (31.3)	<0.001
Vomiting	1958 (12.6)	606 (13.2)	1352 (12.0)	<0.001
Abdominal pain	2114 (13.7)	683 (14.9)	1431 (12.7)	<0.001
Sore throat	5321 (34.4)	1850 (40.4)	3471 (30.8)	<0.001
Conjunctivitis	3074 (19.9)	1104 (24.1)	1970 (17.5)	<0.001
Cyanosis	1703 (11.0)	395 (8.6)	1308 (11.6)	<0.001

The *p*-value corresponds to chi-squared. For the study, the attributes sex, vaccinated for influenza, and diarrhea were excluded with *p*-value > 0.001, and the others were selected.

**Table 2 diagnostics-13-03352-t002:** Results of supervised machine learning algorithms.

Algorithm	AUC	Acc	Rec	Prec	Spec	F1
Random Forest	0.94	0.86	0.91	0.82	0.88	0.86
Bagging	0.93	0.85	0.90	0.82	0.87	0.85
Decision Tree	0.85	0.70	0.71	0.73	0.73	0.72
Kneighbors (7)	0.73	0.63	0.67	0.63	0.60	0.63
Gradient Boosting	0.69	0.62	0.69	0.61	0.56	0.62
SVM rbf	0.67	0.62	0.65	0.61	0.59	0.62
Quadratic Discriminant	0.66	0.62	0.70	0.60	0.54	0.62
Ada Boost	0.66	0.62	0.62	0.61	0.61	0.62
Linear Discriminant *	0.65	0.61	0.62	0.61	0.61	0.61
Linear SVM *	0.65	0.61	0.62	0.61	0.61	0.61
Logistic Regression	0.65	0.61	0.62	0.61	0.61	0.61
BernoulliNB	0.65	0.61	0.59	0.61	0.62	0.61

* Discriminant Analysis and SVM were used twice with different parameters. AUC, area under the curve; Acc, accuracy; Rec, recall; Prec, precision; Spec, specificity; D1, F1-score.

## Data Availability

The data presented in this study are available on request from the corresponding author.

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
