# Peer review of "Supervised Machine Learning Methods for Seasonal Influenza Diagnosis"

_diagnostics, 2023, doi:10.3390/diagnostics13213352_

Round 1

Reviewer 1 Report (Previous Reviewer 1)

Comments and Suggestions for Authors

This study built machine-learning models to discriminate influenza from other diseases based on mainly the symptoms of patients. The reviewer has several concerns about the study.

1 Lots of viruses may cause symptoms of influenza, such as coronaviruses and rhinovirus. There seems to be no influenza-specific symptoms. Thus, it is very challenging to computationally discriminate influenza from other diseases. The excellent performances of random-forest models presented in the manuscript are very likely to be results of over-fitting.

2 Compared to previous version of the manuscript, it is not clear how did the authors improve the ML-modeling. There were no changes on the features and algorithms used, although the data were increased much.

3 There were still lots of grammar errors in the manuscript.

Comments on the Quality of English Language

There were still lots of grammar errors in the manuscript.

Author Response

The authors had reviewed all the sections of the article that you suggested for improvement and have made the necessary changes to enhance them.

Reviewer 2 Report (New Reviewer)

Comments and Suggestions for Authors

Comments and suggestions

Ms. No.: diagnostics-2596727

Title:  Supervised machine learning methods for seasonal influenza diagnosis

Influenza has been a stationary disease in Mexico since 2009 and it represents a high cost for national public health, caused by detection with PCRs, treatments, and absences from work of people. Despite influenza’s relevance, the definition of main features to detect the disease, by international institutions like World Health Organization (WHO) and United States Centers for Disease Control and Prevention Center (CDC), is not a general patron to all people. The aim of this work is to find a machine learning method to support decision-making in the clinical differentiation between positive and negative influenza patients, based on symptoms and demographic features. The research samples consist of records with clinical and demographic information of patients with a positive/negative PCR result of influenza, from 2010 to 2020 in the healthcare institutions of Mexico City. The performance of the methods for classifying influenza cases are evaluated with indices like accuracy, specificity, sensitivity, recall, precision, f-measure and area under curve (AUC). Results indicate that random forest and bagging classifiers were the best supervised methods; they have shown promise in supporting clinical diagnosis, especially in places where performing a PCR test might be challenging or not feasible.

Some comments about this paper:

1)  Why was the Spearman correlation utilized in this study?

2)  The AdaBoost in some dataset was showed the low performance, why did the author use this scheme?

3)  Correct; “test with” not “testwith”

4)  Table 2; Discuss on the obtained results from the algorithm and conditions and the reason for accuracy of the best methods.

5)  Compare the results from some parameters with open literature.

6)  Discuss physically on the obtained results.

 minor revision

Author Response

The authors had reviewed all the sections of the article that you suggested for improvement and  made the necessary changes to enhance them.

Reviewer 3 Report (New Reviewer)

Comments and Suggestions for Authors

Dear Authors,

Congratulations on your outstanding paper, which I had the opportunity to read. You have addressed a significant issue and attempted to present an alternative, cost-effective, and time-saving method for diagnosing Influenza through data management.

I don't have any specific comments to make, but I would like to share an observation. In my opinion, the Discussion section is quite concise compared to the extensive methodological content of the article. While this conciseness is understandable since the paper primarily focuses on statistical methods, it would be interesting if you could briefly mention other medical fields where machine learning models have proven to be valuable diagnostic tools (references 13 to 47). Specifically, I recommend taking a look at the following article:

Altini N, Brunetti A, Mazzoleni S, Moncelli F, Zagaria I, Prencipe B,et al. Predictive Machine Learning Models and Survival Analysis for COVID-19 Prognosis Based on Hematochemical Parameters. Sensors (Basel). 2021 Dec 20;21(24):8503. doi: 10.3390/s21248503. PMID: 34960595; PMCID: PMC8705488.

This article explores the use of machine learning tools to predict the outcomes of patients with COVID-19 and may be a valuable addition to your list of references.

Comments on the Quality of English Language

Good quality of English language. Only moderate grammar and spelling checks are required.

Author Response

The authors had reviewed all the sections of the article that you suggested for improvement and  made the necessary changes to enhance them.

Round 2

Reviewer 1 Report (Previous Reviewer 1)

Comments and Suggestions for Authors

The writing of the manuscript was improved much. However, the reviewer still think that there are no influenza-specific symptoms. As demonstrated by the results, although both RF and bagging algorithms performed excellently on the training dataset, their performances were poor on the testing dataset, with accuracies of about 0.7, suggesting that they are not suitable for diagnosis in applications. 

Comments on the Quality of English Language

The writing was improved much compared to previous version of the mauscript. 

Author Response

Thank you for your comments, they are very important to improve our work and the manuscript.

We had reviewed again the English redaction and made some changes to improve.

We include changes:

"Our results show problems in the prediction of positive influenza cases...

They should not be perceived as a complete substitute for clinical diagnosis. RT-qPCR tests retain their indispensable status for precise influenza results."

We changed our conclusions: "

Machine learning models showcased a notably higher specificity compared to sensitivity, suggesting their potential utility in the identification of negative cases. This capability could help minimize the number of unnecessary molecular tests for individuals presenting with symptoms resembling influenza. This aspect is particularly pertinent in Mexico, where, for epidemiological reasons during the influenza season, around 10% of the population with symptoms resembling influenza is randomly selected for RT-qPCR testing, with approximately 70% of those cases turning out to be negative. By incorporating a tool akin to the one outlined in this study, clinicians can make more informed decisions about which patients require PCR testing, ultimately enhancing data quality for national-level decision-making. Furthermore, given the limited availability of RT-qPCR testing facilities in certain areas, this tool can serve as valuable support for healthcare practitioners, aiding them in determining the necessity of conducting tests. This approach has the potential to reduce costs for patients and ease the burden on the healthcare sector."

"

This manuscript is a resubmission of an earlier submission. The following is a list of the peer review reports and author responses from that submission.

Round 1

Reviewer 1 Report

Comments and Suggestions for Authors

Discrimination of influenza and other diseases is difficult based on the symptoms as there seems to be no influenza-specific symptoms. This study investigated the problem with machine learning methods. Unfortunately, the best model still has poor performance, suggesting that the model cannot be applied in influenza diagnosis. In terms of the manuscript, the Statistical analysis method should be added in the Method section. Besides, there were too many grammar errors in the manuscript.

Reviewer 2 Report

Comments and Suggestions for Authors

I have gone through the study. It's an excellent study of the detection of influenza before the RCT. but I have the following concerns;

1. The abstract should be revised with more explanation about the study and objectivity. it should be divided into different sections like the problem of the study—methodology, results, and conclusions.

2.  different machine learning models have been applied but their definition with some explanation lacking in the study. The algorithms should be explained separately in the methodology section.

3. Is there any previous studies that used the same algorithms? what factors they figured out and what did this study do? which factors were found more significant and contributing towards influenza-positive cases?

4. The study lacks the limitation of the study. And what suggestions does it give for the future policy making.